# Role of HIF-1α in Alcohol-Mediated Multiple Organ Dysfunction

**DOI:** 10.3390/biom8040170

**Published:** 2018-12-10

**Authors:** Niya L. Morris, Samantha M. Yeligar

**Affiliations:** 1Division of Pulmonary, Allergy, Critical Care and Sleep Medicine, Department of Medicine, Emory University, Atlanta, GA 30322, USA; 2Division of Pulmonary, Allergy, Critical Care and Sleep Medicine, Department of Medicine, Atlanta Veterans Affairs Health Care System, Decatur, GA 30033, USA; niya.morris@emory.edu

**Keywords:** ethanol, alcohol, HIF-1α

## Abstract

Excess alcohol consumption is a global crisis contributing to over 3 million alcohol-related deaths per year worldwide and economic costs exceeding $200 billion dollars, which include productivity losses, healthcare, and other effects (e.g., property damages). Both clinical and experimental models have shown that excessive alcohol consumption results in multiple organ injury. Although alcohol metabolism occurs primarily in the liver, alcohol exposure can lead to pathophysiological conditions in multiple organs and tissues, including the brain, lungs, adipose, liver, and intestines. Understanding the mechanisms by which alcohol-mediated organ dysfunction occurs could help to identify new therapeutic approaches to mitigate the detrimental effects of alcohol misuse. Hypoxia-inducible factor (HIF)-1 is a transcription factor comprised of HIF-1α and HIF-1β subunits that play a critical role in alcohol-mediated organ dysfunction. This review provides a comprehensive analysis of recent studies examining the relationship between HIF-1α and alcohol consumption as it relates to multiple organ injury and potential therapies to mitigate alcohol’s effects.

## 1. Introduction

Alcohol is one of the most abused substances in the United States, claiming the lives of nearly 100,000 individuals within the United States and accounting for 3.3 million alcohol-related deaths worldwide annually [1,2,3,4]. Alcohol misuse is the third leading cause of preventable death, with economic costs exceeding $200 billion dollars due to healthcare costs, productivity losses, and other effects of alcohol use (e.g., property damages) [1,3,5]. The impact of alcohol is multifactorial, as it increases the risk of alcohol-related morbidity and mortality [6,7]. Even though the liver is the major site of alcohol metabolism, other organs and tissues, including the brain, lungs, adipose, and intestines, are impacted by alcohol exposure [8,9,10,11]. Alcohol consumption has been associated with increased risk of Parkinson’s disease, acute respiratory distress syndrome, diabetes, liver cirrhosis, and colon cancer development [6,12,13]. Therefore, determining the mechanisms involved in the debilitating effects of alcohol exposure on specific organs and tissues is imperative to potentially identify novel therapeutic interventions.

## 2. Molecular Mechanisms Which Regulate HIF Activity

Hypoxia-inducible factor (HIF)-1 is a heterodimeric transcription factor, composed of HIF-1α and HIF-1β subunits. The chief function of this complex is to allow for cellular adaptation. The HIF-1α subunit is constitutively expressed [14]. However, in normoxia, the HIF-1α protein is post-translationally modified on the oxygen-dependent degradation domain of the protein via hydroxylation and acetylation by prolyl hydroxylases (PHDs) and arrest-defective-1 respectively [14]. PHDs use the substrates oxygen and iron as cofactors which lead to hydroxylation of the prolines on HIF-1α, resulting in HIF-1α ubiquitination by a von Hippel Lindau E3 ubiquitin ligase (VHL) and its subsequent degradation [14,15]. Under hypoxic conditions, the HIF-1α subunit does not undergo post-translational modifications by prolyl hydroxylases, which allows HIF-1α to translocate to the nucleus and bind to HIF-1β, forming the heterodimer HIF-1 [14].

In addition to this oxygen-dependent regulation of HIF-1α stabilization, there are also oxygen-independent mechanisms of HIF-1α stabilization by reactive oxygen species (ROS) and inflammatory mediators. There is some debate about the exact role of ROS in HIF-1α stabilization and no definitive conclusions have been established. Some studies have reported that elevated ROS levels result in either diminished HIF-1 DNA binding or increased degradation of HIF-1α [16,17], whereas other studies have shown that hormone-induced ROS generation and exogenous ROS treatment regulate HIF-1α stability by specific signaling pathways (e.g., phosphatidylinositol-3-kinase (PI-3K)), inhibition of HIF-1α/VHL binding, and diminished PHD activity [18,19,20,21,22]. Similarly, inflammatory mediators (such as nuclear factor kappa-light-chain-enhancer of activated B cells (NF-κB), tumor necrosis factor (TNF)-α, transforming growth factor (TGF)-β1 and interleukin (IL)-1β) have been shown to regulate HIF-1α transcription and stabilization by reducing PHD expression, and the PI-3K/mammalian target of rapamycin pathways [23,24,25,26]. The regulation of HIF-1α is particularly important as the HIF-1 complex (comprising the stabilized HIF-1α and its binding partner HIF-1β) acts as a transcription factor to regulate hundreds of genes [27,28]. Genes that are regulated by HIF-1 are involved in processes such as angiogenesis, oxidative stress, cell proliferation, and apoptosis [14]. Interestingly, environmental factors such as alcohol exposure can elevate ROS and/or inflammation and induce HIF-1α activation.

Numerous studies have shown a direct relationship between HIF-1α and alcohol-mediated pathologies. Understanding the mechanisms by which alcohol exposure results in organ damage (e.g. HIF-1α expression) could result in potential therapeutic strategies to mitigate the adverse effects associated with alcohol use. Time of exposure (prenatal vs. adulthood) and type of exposure (acute vs. chronic) appear to be important in determining the expression pattern of HIF-1α. Treatment options to mitigate HIF-1α effects range from antioxidants, microRNAs, probiotics, and HIF-1α pharmacological treatments. The current review outlines recent studies examining the role of HIF-1α in alcohol-mediated multiple organ injury and current therapeutic approaches to minimize alcohol-related pathologies.

## 3. The Role of HIF-1α in Alcohol-Mediated Brain Damage

As alcohol is one of the select molecules with the ability to cross the blood–brain barrier, it is unsurprising that there are alcohol-mediated effects on the brain [29]. Both clinical and experimental models have demonstrated that alcohol impairs brain structure and function [30,31]. Alcohol has also been shown to weaken the integrity of the blood–brain barrier [31]. These adverse effects likely contribute to the clinical observations that alcohol increases the risks of developing dementia, Parkinson’s disease, and Alzheimer’s disease [12].

The effects of HIF-1α on alcohol-mediated brain damage appear to be dependent on when alcohol exposure takes place (fetal exposure vs. adult exposure) and the duration of alcohol exposure (acute, binge, or chronic). Maternal alcohol ingestion during gestation results in fetal alcohol spectrum disorders, which are associated with impaired fetal development. Prenatal alcohol exposure is associated with teratogenesis, including deficient development of motor skills and developmental growth derangements [32,33], as evidenced by reduced heart, limb, and brain development [32]. Chronic prenatal alcohol exposure in rats significantly reduced HIF-1α levels [33,34], which is likely due to the decreased expression of HIF-1α stabilization proteins, such as heat shock protein (HSP)-70 [34]. Tong et al. investigated the role of prenatal exposure of ethanol on juvenile rats, which impaired the insulin/insulin-like growth factor (IGF) pathway [33]. Since both insulin and IGF-1 have been shown in other models to stabilize HIF-1α in mouse preadipocytic cells and in human retinal epithelial cells [22,35], diminished insulin and IGF following prenatal alcohol exposure may attribute to the downregulation of HIF-1α seen in the brain by similar mechanisms. Similar to prenatal alcohol exposure [33,34], chronic binge alcohol exposure decreased HIF-1α expression [36]. Interestingly, this study also demonstrated reductions in the insulin/IGF-1-regulated gene aspartyl–asparaginyl–β-hydroxylase. This study suggests that chronic binge alcohol patterns produce similar impairments of the insulin/IGF pathway [36] as that of prenatal alcohol exposure [33]. Since HIF-1α is critical for angiogenesis, diminished HIF-1α levels caused by prenatal alcohol exposure could be detrimental to the developing embryo. Similarly, an adult rat ischemic stroke model demonstrated that acute alcohol exposure was neuroprotective in reducing infarct volume and improving motor skills due to acute alcohol-induced HIF-1α expression [37]; this is likely due to its ability to increase angiogenesis.

By contrast, Reddy et al. described how chronic alcohol exposure in adult rats increased HIF-1α mRNA and protein in the brain cortex [38]. The group speculated that the observed elevation in HIF-1α resulted from a hypoxic insult, which likely occurred due to increased oxygen uptake following alcohol metabolism [38]. Interestingly, the observed elevated HIF-1α was associated with increased oxidative stress [18,19,20,21,22,38]. Further, chronic alcohol consumption in rats increased mitochondrial dysfunction and mitochondrial lipid peroxidation while decreasing activities of mitochondria complexes I, III, and IV. Nitrosative stress is induced following chronic alcohol administration via elevated nitrite/nitrate levels, inducible nitric oxide synthase (iNOS), and neuronal nitric oxide synthase (nNOS) in the brain cortex. Further, alcohol decreased the mRNA and protein levels of mitochondrial specific antioxidant superoxide dismutase (SOD)2 [38]. In addition to ROS, inflammation also elevates HIF-1α levels [23,24,25,26]. Models of acute ethanol exposure and traumatic brain injury(TBI) demonstrate a similar elevation of HIF-1α [39,40], which are likely due to the observed increase in expression of NF-κB following acute ethanol and TBI injury [39]. Similar responses were observed in mice pretreated with an oxidative stressor, DL-buthionine-(S, R-sulfoximine) [39]. These data suggest that the response is mediated by oxidative stress-induced HIF-1α; however, alcohol could be simply exacerbating the hypoxic insult or the inflammatory response in the brain following TBI, resulting in increased HIF-1α stabilization [41]. Additional studies are needed to definitively determine the mechanism by which alcohol induces HIF-1α following TBI.

These findings demonstrate that alcohol exposure influences the expression of HIF-1α in the brain. Interestingly, these studies suggest that the time and type of alcohol exposure appear to influence the effects of HIF-1α on alcohol-mediated brain damage. As the transcription factor HIF-1 regulates the expression of hundreds of genes [27,28], this is likely due to the needs of the organ (e.g., angiogenesis) at the time of alcohol exposure. Due to the downstream responses of HIF-1α stabilization, it is unsurprising that a lack of its stabilization would have different functional consequences on alcohol-mediated brain injury.

## 4. The Role of HIF-1α in Alcohol-Mediated Lung Damage

Similar to what is observed in the brain, prenatal alcohol exposure negatively impacts fetal lung development [42]. Preterm animals born to pregnant ewes that were fed alcohol during the last trimester of their pregnancy showed reduced expression of HIF-1α, vascular endothelial growth factor (VEGF)-α, and VEGF receptor (VEGFR)-1. Additionally, prenatal alcohol exposure reduced lung protein levels and diminished the levels of TNF-α, IL-10, monocyte chemotactic protein (MCP)-1, and C–C motif ligand (CCL)-5 in preterm animals [42]. As inflammation has been shown to stabilize HIF-1α [23,24,25,26], reduced levels of these pro-inflammatory mediators likely resulted in reduced HIF-1α expression in this model of prenatal alcohol exposure. More studies are needed to adequately address the role of HIF-1α on alcohol-mediated lung injury.

Both clinical and experimental studies have described the negative impacts of alcohol exposure on lung function [43,44]. Chronic alcohol abuse increases the risk of developing respiratory infections [45], which is associated with alveolar macrophage (AM) phagocytic dysfunction [44,46,47]. Suppression of the AM function is particularly problematic, as AMs initiate the immune response to invading pathogens [48]. Our laboratory has shown that alcohol-mediated AM phagocytic dysfunction is largely linked to reduced peroxisome proliferator-activated receptor (PPAR)γ [44]. Additionally, we observed that this coincided with an increased expression of TGF-β_1_ [44]. Studies in an in vitro pulmonary hypertension model have demonstrated that treatment with the PPARγ ligand rosiglitazone downregulated HIF-1α expression [49]. As described earlier, increased ROS has been linked to elevated HIF-1α levels. Interestingly, chronic alcohol exposure increases oxidative stress in AMs [43,44,50]. Reduced HIF-1α following treatment with the PPARγ ligand could be due to the ability of PPARγ to reduce oxidative stress [44]; however, the direct relationship between PPARγ and HIF-1α specifically following chronic alcohol exposure needs further exploration. In addition to increased ROS, TGF-β_1_ is also increased following chronic alcohol exposure [44]. It has been shown in other experimental models that TGF-β_1_ regulates the levels of HIF-1α [23].

Collectively, these findings demonstrate that alcohol ingestion impairs lung function. Lazic et al. showed that preterm alcohol exposure diminished the expression of HIF-1α by unknown mechanisms. One potential mechanism is the observed reduction in inflammation following prenatal alcohol exposure, which could result in an inability to stabilize HIF-1α [42]. Additional studies are needed to definitively determine the mechanism behind the reduction of HIF-1α in the lungs due to prenatal alcohol exposure. There is limited knowledge regarding the role of alcohol on HIF-1α expression and alcohol-mediated lung injury. Although it is widely known that alcohol impairs lung and/or alveolar macrophage dysfunction, which is largely due to increased oxidative stress [43,44,46,47,50], further research is needed to elucidate the interactions between oxidative stress and HIF-1α following acute or chronic alcohol exposure.

## 5. The Role of HIF-1α in Alcohol-Mediated Adipose Damage

Alcohol abuse has been associated with pathophysiological conditions that are affected by metabolism, such as diabetes [12]. Chronic alcohol exposure impacts lipolysis and lipogenesis balance, as demonstrated by reduced fat mass in alcoholic individuals compared to control subjects [51]. Similarly, chronic alcohol consumption reduced adipocyte size and tissue mass in mice [52]. These studies demonstrate that alcohol can decrease fat mass, likely due to alterations in lipid oxidation. By contrast, alcohol-treated rats had larger epidydimal and perirenal adipose cells and lipid droplet sizes compared to control animals, which suggests that ethanol increases adipocyte differentiation and lipid accumulation [53]. Alcohol also impacts glucose tolerance and sensitivity to insulin. Alcohol-fed animals had elevated blood glucose levels thirty minutes after receiving an intraperitoneal injection of insulin [52]. HIF-1α can influence both lipid and glucose metabolism; the administration of an HIF-1α antisense oligonucleotide decreased fasting blood glucose levels and lipid biosynthesis [54]. Chronic alcohol exposure elevated HIF-1α and its downstream target glucose transporter (GLUT)1 in the epidydimal adipose tissue of Wistar rats compared to controls and OP9 adipocytes. Further, alcohol reduced glucose tolerance and increased the levels of TNF-α, IL-6, VEGF, and leptin [53]. Interestingly, one of the main characteristics of chronic and acute alcohol consumption is increased hepatic triglycerides [55], which are transported from adipose tissues to the liver and contribute to alcohol-induced fatty liver [52]. The mechanisms by which ethanol consumption results in elevated HIF-1α in adipose tissue have not been definitively concluded. However, the authors speculate that based on the elevated levels of HIF-1α and GLUT1 along with visceral adipose tissue mass enlargement, long term alcohol exposure may have resulted in a hypoxic insult in the adipocytes [53]. Collectively, the data demonstrate that alcohol consumption results in increased inflammation and HIF-1α expression, which provides evidence into a possible mechanism and potential treatment target for ethanol-related metabolic conditions, such as ethanol-related diabetes.

## 6. The Role of HIF-1α in Alcohol-Mediated Liver Damage

The liver is the primary organ of alcohol metabolism [8]. Clinical and animal models of alcohol exposure show elevations in plasma alanine aminotransferase (ALT) and/or lipopolysaccharides (LPS)/endotoxin levels [55,56,57,58,59,60,61,62,63,64,65,66,67]. Alcohol feeding increases liver steatosis, lipid peroxidation, and/or hepatic damage [55,56,57,58,59,61,63,64,66,68,69,70,71,72]. Chronic alcohol exposure also decreases mitochondrial respiratory chain protein activity [70]. Additionally, alcohol consumption increases liver hypoxia [58,68,69], ROS [55,58,60,63,64,67,70], and inflammation [58,59,60,61,63,66,67,72], which is due in part to the enhanced levels of pro-inflammatory mediators in endothelial cells [73,74] and Kupffer cells [75].

The relationship between HIF-1α and alcohol consumption has been widely examined in the liver. Both binge and chronic alcohol consumption elevated HIF-1α in the liver [55,58,59,60,61,62,63,64,66,67,68,70,71,72], including in endothelial cells [73,74,76] and Kupffer cells [77]. Mice with hepatocyte-specific knockout of HIF-1α demonstrate reduced inflammation, steatosis, and serum ALT following chronic alcohol feeding [61,66]. Conversely, Nishiyama et al. found that hepatocyte-specific deletion of HIF-1α had deleterious effects, including increased expression of lipogenic genes [68], suggesting that HIF-1α plays a critical role in hepatocytes during chronic alcohol ingestion.

Cytochrome P450 (CYP2E1) is an enzyme that is required for alcohol metabolism [8]. CYP2E1 is increased in mice in acute, binge, and chronic alcohol exposure models and in alcoholic patients [58,70]. Increases in CYP2E1 positively correlated with HIF-1α and blood alcohol concentration in mice exposed to binge alcohol and in alcoholic patients [58]. CYP2E1 knockout mice have reduced alcohol-mediated pathology. On the other hand, CYP2E1 knock-in mice exposed to a chronic alcohol model exhibited exacerbated liver injury, characterized by increased serum ALT levels, lipid peroxidation, hypoxia, inflammatory cell infiltration, CYP2E1 activity, cell degeneration, oxidative stress, and HIF-1α. Further, CYP2E1 knock-in mice had reduced mitochondrial glutathione (GSH) compared to their wild type and CYP2E1 knockout counterparts [69].

Two hit models have demonstrated that alcohol compounds injury [78,79]. In a model of diethyl–nitrosamine -induced hepatobiliary cancer, alcohol feeding increased serum ALT, liver inflammation, hepatobiliary cyst formation, proliferation (BrdU, p53 and cyclin D1), and upregulated cancer stem cell markers (nanog and CD133). These phenotypic changes correlated with a decrease in microRNA (miR)-122 and upregulation of its target HIF-1α [60]. Similarly, reductions of miR-122 were seen in patients with alcoholic cirrhosis and in mice given the Lieber DeCarli ethanol feeding diet compared to controls. Downregulation of miR-122 is due to an elevated expression of a spliced form of grainyhead-like transcription factor 2 which also resulted in enhanced HIF-1α levels [66]. Chronic alcohol exposure also reduced the expression of miR-199 in sinusoidal endothelial cells, which contributes to alcohol-induced HIF-1α and subsequent elevation of vasoconstrictor endothelin-1 [73]. These studies demonstrate that alcohol impacts the expression of HIF-1α-related miRs and they are, therefore, excellent candidates for therapeutic treatments for alcohol-induced organ injury.

These studies collectively show that HIF-1α is elevated in the liver following alcohol exposure regardless of the type of exposure. This could be due to the fact that the liver is the primary site of alcohol metabolism; however, more investigation is needed to adequately define this relationship. The data do, however, demonstrate a role of HIF-1α in alcohol-induced liver damage.

## 7. The Role of HIF-1α in Alcohol-Mediated Intestinal Damage

The gut is one of the largest bacterial reservoirs within the body, and the breakdown of this barrier likely contributes to numerous pathological conditions following alcohol exposure [13,80]. Alcohol alone or in combination with injury/disease results in impaired intestinal barrier integrity [81,82,83,84,85,86,87,88,89]. Both binge and chronic alcohol feeding impaired intestinal permeability and the expression of intestinal tight junction proteins (Zonula occludens (ZO)-1, claudin-1, and occludin) [56,57,90], reduced barrier protective molecules such as intestinal trefoil factor (ITF) [56,57,90], and increased levels of reactive oxygen formation in the gut [56,57].

As HIF-1α is largely impacted by ROS levels, investigating the role of HIF-1α in the gut following alcohol injury is extremely important. Chronic and binge alcohol exposure in the gut resulted in differential patterns of HIF-1α expression. Chronic alcohol consumption reduced the expression of both mRNA and protein for HIF-1α [90], whereas binge alcohol exposure did not affect HIF-1α mRNA but decreased the HIF downstream target VEGF [57].

Although there are conflicting roles of HIF-1α expression presented in recent literature regarding chronic vs. binge alcohol exposure, studies of both models highlight the importance of the HIF pathway in alcohol-mediated intestinal pathology. An intestinal epithelial cell knockout of HIF-1α in mice reduced survival following alcohol exposure, and these mice demonstrated increased neutrophil infiltration, macrophage activation, inflammation, serum ALT levels, and enhanced liver steatosis. Chronic alcohol exposure resulted in gut dysbiosis, exhibiting reduced levels of *Lactobacillus* spp. A knockout of HIF-1α exacerbated gut dysbiosis, where the mice showed increased *Bacteroidetes* and *Akkermansia* spp and decreased *Firmicutes*, but they also had nearly undetectable levels of *Lactobacillus* spp. [90].

Both binge and chronic alcohol exposure affect antimicrobial peptides. Chronic alcohol exposure increased antimicrobial peptides (Defensin (Def)β1 and Defβ2) [90], while both binge and chronic alcohol consumption decreased the expression of cathelicidin-related antimicrobial peptides (CRAMP) [56,90]. Chronic alcohol exposure in HIF-1α knockout mice failed to increase Defβ1 and Defβ2 and further depressed expression of CRAMP and tight junction proteins (occludin, claudin-1) [90]. Many of the adverse effects associated with binge and chronic alcohol exposure are ameliorated when treated with agents that increase HIF-1α expression [56,57,90], illustrating that HIF signaling has a role in gut barrier maintenance following alcohol exposure.

Numerous studies demonstrate that alcohol exacerbates adverse effects to the gut following injury and disease [82,87,91,92,93]. In a two-hit model of co-exposure of colon cancer cell lines to alcohol and/or arsenic, alcohol or arsenic alone increased ROS production; however, there were additive effects when alcohol and arsenic exposure were combined. These effects were attenuated when cells were treated with antioxidants. Following arsenic and alcohol exposure, similar increases were found in the expression and secretion of HIF-1α and its downstream target VEGF, which plays a role in tumorigenesis. The use of antioxidants (catalase, SOD, manganese (III) tetrakis (4-benzoic acid) porphyrin chloride (MnTBAP)) or an HIF-1α inhibitor LW6 attenuated alcohol- and arsenic-induced vascular endothelial cell tube formation, suggesting that ROS-induced HIF-1α signaling is instrumental in alcohol/arsenic-induced tumorigenesis [94].

Similar to the brain, the effects on HIF-1α expression following alcohol exposure appear to be dependent on the type of alcohol exposure (acute vs. binge/chronic). Collectively, these studies demonstrate that alcohol exposure impacts HIF-1α expression and/or stabilization, which can lead to detrimental effects in the intestine (impaired intestinal permeability and gut dysbiosis), and that treatment with agents to mitigate alcohol’s effects on HIF-1α could lead to therapeutic strategies to reduce alcohol pathophysiology.

## 8. Treatment Options for Alcohol-Mediated Tissue/Organ Damage

### 8.1. Antioxidants

HIF-1α is regulated by proline hydroxylation by PHDs, which use iron as a cofactor, which results in hydroxylation and its ubiquitination [14,15]. Some studies have illustrated that ROS has the ability to regulate HIF-1α stability, although this is still under debate [18,19,20,21,22,23]. Further, numerous studies have demonstrated that alcohol exposure elevates ROS [55,56,57,60,63,64,67,70] and reduces hepcidin [71,95], which is a liver-derived regulator of iron homeostasis. Similarly, alcohol can alter the expression of antioxidant proteins, such as heme oxygenase-1 [77,96] and NAD(P)H:quinone oxidoreductase [59,77]. Therefore, antioxidants have been used in many studies examining potential therapeutic approaches for alcohol-induced organ injuries.

The antioxidant M30 (5-[N-methyl[N[propargylaminomethyl]-8-hydroxy-quinoline) is an iron chelator and has been investigated for its ability to attenuate alcohol-induced injury in hepatocytes. Pretreatment with M30 partially reduced apoptosis, ROS levels, and the secretion of pro-inflammatory mediators (TNF-α, IL-6, IL-1β and IL-18) in hepatocytes. M30 also elevated the levels of antioxidants, including catalase and GPx1. Furthermore, M30 treatment diminished the adenylate cyclase/cyclic adenosine monophosphate/protein kinase A/HIF-1α pathway. Similar changes were observed using CAY10585, an HIF-1α inhibitor, suggesting that the protective effect of M30 to alcohol-induced injury in hepatocytes is due to its ability to reduce the levels of HIF-1α [67].

Resveratrol (RES) is a natural polyphenol that has antioxidant properties and inhibits NADPH oxidases. RES significantly diminished liver fat deposition, serum ALT and aspartate aminotransferase (AST), liver HIF-1α protein levels, and mitochondrial ROS production in ethanol-fed rats [64]. Vitamin C is another antioxidant that has been shown to have protective effects against alcoholic liver disease by suppressing HIF-1α stabilization [62]. Similarly, treatment with the mitochondria-target antioxidant ubiquinone (MitoQ) reduced alcohol-induced HIF-1α expression, hepatic steatosis, lipid peroxidation, and protein nitration in rats [70].

Ma et al. demonstrated that pre-endurance training (12-week exercise training) minimized the effects of a 5-day alcohol (5 g/kg) exposure in the liver. This pre-endurance exercise training reduced plasma ALT and AST levels, liver mitochondrial ROS production, lipid peroxidation (TBARS), and the NADH/NAD+ ratio. Pre-endurance training also reduced the expression of HIF-1α mRNA and protein, suggesting that since exercise increases antioxidant production, it can be used as a therapy for alcohol exposure [55]. Collectively, these studies provide evidence into the relationship between HIF-1α and ROS-mediated stabilization. Further, they illustrate that antioxidant treatment and subsequent attenuation of alcohol-induced HIF-1α expression can mitigate the adverse effects following alcohol consumption.

### 8.2. microRNAs

Beyond alcohol affecting transcription factors such as HIF-1, alcohol also modulates the expression of microRNAs (miRs) [60,66,73,83,84]. miRs are predicted to regulate 60% of the genome [97,98], and altered expression of these molecules can affect thousands of genes. Studies are now being proposed to treat with microRNAs to examine whether they can improve alcohol-mediated injuries. HIF-1α is a direct target of miR-122; therefore, reductions in miR-122 led to elevated HIF-1α levels [60,66]. Since miR-122 is significantly reduced following chronic alcohol exposure, pretreatment of mice with an injection of a precursor for miR-122 showed reductions in alcohol- and carbon tetrachloride -induced serum ALT levels and liver inflammation (MCP-1 and IL-1β), fibrosis, and steatosis, compared to controls. The relationship between miR-122 and HIF-1α suggests that pretreatment with the miR-122 precursor could attenuate alcohol-induced HIF-1α levels. These data demonstrate that in addition to possibly being of use as biomarkers, miRs can also be used for therapeutic interventions to treat alcohol-mediated pathologies.

### 8.3. Probiotics

Alcohol exposure alone has been shown to result in dysbiosis of the gut microbiome; in particular, *Lactobacillus* levels are reduced following alcohol exposure [90]. Treatment with probiotics could restore intestinal barrier integrity following alcohol exposure. *Lactobacillus rhamnosus* Gorbach Goldin (LGG) pretreatment reduced liver steatosis and triglyceride levels, plasma ALT, LPS, and liver triglyceride levels in chronic and binge alcohol models [56,57]. The effects of LGG are mediated by its ability to restore VEGF and intestinal trefoil factor expression. LGG treatment elevated HIF-1α mRNA levels following binge alcohol consumption [56]. Further, LGG attenuated alcohol-mediated increases in ROS and decreases in tight junction proteins (ZO-1, claudin-1, and occludin) while improving paracellular permeability following both binge and chronic models of alcohol exposure [56,57]. LGG supplementation also increased the expression of barrier-protecting molecules, such as CRAMP, permeability glycoprotein, and ITF, which helps form the mucus layer following alcohol exposure [56,57]. A siRNA-mediated silencing of HIF-1α/2α prevented LGG-mediated protective effects against alcohol [57]. The use of antibiotics reduced liver fat accumulation and serum ALT levels, illustrating that gut dysbiosis is necessary for liver damage and steatosis. The use of intestinal epithelial specific HIF-1α knockout mice treated with LGG in the presence of chronic alcohol exposure ameliorated the therapeutic effects of LGG [90]. These studies suggest that probiotics such as LGG work on the HIF pathway to provide its therapeutic effects following alcohol exposure [56,57,90].

### 8.4. Pharmacological Treatments

Pharmacological treatments have also been utilized to improve alcohol-induced pathologies. In conditions in which diminished HIF-1α levels are associated with damage, e.g., brain [34], prenatal exposure to ethanol and intraperitoneal treatment with tiocetam (125 mg/kg) and cerebrocurin (0.06 mg/kg) increased HIF-1α and HSP-70 levels and was neuroprotective against alcoholic encephalopathy [34]. Similarly, prenatal exposure and treatment with emodin increased HIF-1α, improved embryo development, and increased the levels of antioxidants SOD1 and SOD2 while decreasing inflammation (TNF-α) and apoptosis as measured by caspase 3 levels [32]. Interestingly, the diminished levels of HIF-1α, inflammation, and increased levels of the antioxidants provide further evidence into the relationship between ROS and HIF-1α stabilization.

Another potential treatment is Ligustrazine, which is the bioactive component of *Ligusticum chuanxiong* Hort and activates nuclear factor erythroid derived 2 like 2 (Nrf2). Nrf2 is an antioxidant regulator which is reduced following alcohol exposure. Although Nrf-2 knockdown exacerbated alcohol-induced plasma ALT, AST, and alkaline phosphatase (ALP) levels in mice, treatment with ligustrazine reduced plasma fatty acids and activities of AST, ALT, and ALP. Ligustrazine also reduced liver inflammation and increased PPAR-α following chronic alcohol exposure. A knockdown of Nrf-2 in the presence of ligustrazine or treatment with the HIF-1α agonist dimethyloxalylglycine failed to demonstrate the protective effects of ligustrazine. Ligustrazine reduced HIF-1α levels, and HIF-1α siRNA showed similar beneficial effects to that of ligustrazine, illustrating the role of HIF-1α in alcohol-mediated organ injury [59].

PX-478, an HIF-1α inhibitor, has been utilized in treating binge alcohol-induced liver damage. Pretreatment of mice with PX-478 partially reduced HIF-1α and BCL2 interacting protein 3 expression, iNOS, and plasma ALT levels. PX-478 pretreatment trended to reduce hepatocyte apoptosis in binge alcohol mice. These data suggest that HIF-1α plays an important role in binge alcohol-induced hepatotoxicity [58]. Similarly, methoxyestradiol has also been used as an HIF-1α pharmacological inhibitor in E47 cells (which constitutively express CYP2E1). Methoxyestradiol treatment attenuated the adverse effects of alcohol and arachidonic acid or alcohol and buthionine sulfoximine. Inhibition of HIF-1α blocked ethanol/arachidonic acid and alcohol/buthionine sulfoximine elevation of oxidative stress and increased toxicity [69]. Together, these studies provide evidence of the relationship between HIF-1α and CYP2E1 in liver injury due to alcohol exposure.

Collectively, these studies illustrate the necessity to understand the effects of alcohol on HIF-1α. HIF-1 regulates hundreds of genes [27,28]; therefore, altered expression, whether an increased or decreased expression of the stabilized subunit HIF-1α, could have widespread effects on gene expression of its targets. For example, in instances in which ethanol reduced the expression of HIF-1α [32,34], agents which elevated HIF-1α aided in alleviating alcohol-mediated adverse effects, whereas in circumstances in which alcohol-induced HIF-1α expression, the use of treatments that knocked down HIF-1α improved outcomes associated with alcohol exposure. Therefore, understanding the HIF-1α needs of the organ following alcohol exposure is required to develop the appropriate therapeutic interventions.

## 9. Conclusions

HIF-1α is a critical molecule of interest in understanding alcohol-mediated organ injury, as alcohol exposure has been shown to alter its expression. The studies outlined here clearly show that differential expression of HIF-1α is due to the type (acute/binge/chronic) and time of alcohol exposure (prenatal vs. adult) (Figure 1). As HIF-1α is a component of the transcription factor HIF-1, which regulates hundreds of genes [27,28], changes in its expression can have varying consequences on organ injury. The consequences of alcohol-mediated changes in HIF-1α are likely dependent on the needs of the organ following alcohol exposure. For example, elevated HIF-1α expression was determined to be neuroprotective following an adult rat ischemic stroke model [37], which is likely due to the ability of HIF-1 to increase angiogenesis, while alcohol-induced HIF-1α was determined to be detrimental in the brain cortex, as it was associated with oxidative stress [38].

The studies outlined here clearly demonstrate that the impact of alcohol exposure is multifactorial and has varying effects on HIF-1α stabilization in different organs. In the brain, alcohol’s impact on HIF-1α appears to be dependent on the time (prenatal vs. adult) and type of alcohol exposure. Limited studies have evaluated the relationship between alcohol and HIF-1α stabilization in the context of the lungs and adipose tissue. The one study evaluating HIF-1α expression following alcohol exposure demonstrated that prenatal alcohol exposure reduced HIF-1α expression in the lungs, which is consistent to what is observed in the brain. In adipose tissue, the study highlighted in this review showed that chronic alcohol exposure elevated HIF-1α in adipose tissue, which could provide insight into how alcohol contributes to metabolic conditions such as alcohol-related diabetes. Further, the studies outlined here illustrate that HIF-1α is elevated in the liver following alcohol exposure independent of the type of alcohol exposure and contributes to alcohol-induced liver injury. In the intestines, the studies reviewed here illustrate that HIF-1α stabilization appears to be dependent on the type of exposure (chronic vs. acute) and that the alcohol-mediated effects on HIF-1α can influence gut dysbiosis and intestinal permeability. These studies illustrate that HIF-1α is a multifaceted molecule that can have differential effects on alcohol-mediated organ injury and that there is no clear consensus on the exact role of HIF-1 (protective vs. harmful) following alcohol-mediated injury. Treatment options such as antioxidants, microRNAs, probiotics, and pharmacological treatments (Table 1) may provide new therapeutic avenues to the negative effects of alcohol exposure as it relates to alcohol-mediated induction or depression of HIF-1α. Additional research is needed to delineate the organ specific changes HIF-1α following different types and timing of alcohol exposure, which can result in the development of novel therapeutic interventions.

## Figures and Tables

**Figure 1 biomolecules-08-00170-f001:**
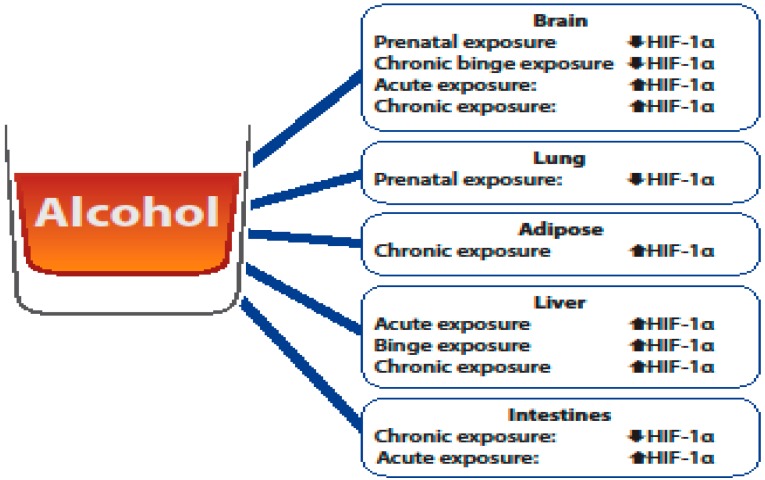
Overview of the differential effects of alcohol of hypoxia-inducible factor (HIF)-1α levels.

**Table 1 biomolecules-08-00170-t001:** Treatment options for alcohol-mediated tissue/organ damage.

Treatment Type	Effects on HIF-1α	Source
**Antioxidants**		
M30	Decreased HIF-1α	Xiao et al. Oxid Med Cell Longev, 2015.
Resveratrol	Decreased HIF-1α	Ma et al. PLoS One, 2017.
Vitamin C	Decreased HIF-1α	Guo et al. Acta Biochimica et Biophysica Sinic, 2013.
MitoQ	Decreased HIF-1α	Chacko et al. Hepatology, 2011.
Pre-endurance Training	Decreased HIF-1α	Ma et al. Hepatol Int, 2014.
**microRNAs**		
miR-122	Decreased HIF-1α	Satishchandran et al. Gastroenterology, 2018.
**Probiotics**		
*Lactobacillus rhamnosus GG*	Increased HIF-1α	Wang et al. The American Journal of Pathology, 2012.
**Pharmacological Treatments**		
Tiocetam	Increased HIF-1α	Belnichev et al. Klinik Psikofarmakoloji Bülteni-Bulletin of Clinical Psychopharmacology,2016.
Cerebrocurin	Increased HIF-1α	Belnichev et al. Klinik Psikofarmakoloji Bülteni-Bulletin of Clinical Psychopharmacology,2016.
Emodin	Increased HIF-1α	Yon et al. Birth Defects Res B Dev Reprod Toxicol, 2013.
Ligustrazine	Decreased HIF-1α	Lu et al. Toxicol Sci, 2017.
PX-478	Decreased HIF-1α	Yun et al. Free radical biology & Medicine, 2014.
Methoxyestradiol	Decreased HIF-1α	Wang et al. Free Radic Biol Med, 2013.

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
