# Peer review of "Role of HIF-1α in Alcohol-Mediated Multiple Organ Dysfunction"

_biomolecules, 2018, doi:10.3390/biom8040170_

Round 1

Reviewer 1 Report

In this review manuscript, the contents had summarized relevant findings of HIF-1a in alcohol-induced tissue/organ injury, including brain, lung, adipose, liver, and intestine. Moreover, several treatment options had been proposed. Basically, the information provided valuable backgrounds and insights into the role of HIF-1a on alcohol-induced diseases. I had only one minor suggestion for the improvement of comprehension. Summarized figures or tables of HIF-1a axes in alcohol-induced tissue/organ injury and treatment options will be helpful to the readers.

Author Response

Response 1) We have added a figure to summarize the effects of HIF-1α in alcohol-induced tissue/organ injury (Figure 1) and a table outlining potential treatment options (Table 1).

Reviewer 2 Report

I read the manuscript entitled "Role of HIF-1α in Alcohol-Mediated Multiple Organ Dysfunction" submitted to the biomolecules as a review article.

In this review, the authors will provide a comprehensive analysis of recent studies examining the relationship between HIF-1α and alcohol consumption as it relates to multiple organ injury and potential therapies to mitigate alcohol’s effects.

I think this review may attract general interests from broad readers.

However, I also think that there are several issues to be approached by the authors to improve the manuscript.

Followings are my comments to the manuscript.

# Molecular mechanisms of regulation of HIF activity

I think that a chapter of molecular mechanisms of regulation of HIF activity can provide more precise information to potential readers.

Although HIFs are identified as the transcription factor activated in response to hypoxia, those can be activated by various stimulation or stress including inflammation and reactive oxygen species (ROS) by the different molecular mechanism from that in the case of hypoxia.

# Lacks of proper citations

Not a small number of statement lacks proper citations throughout the manuscript.

For example, in page 2 "Alcohol increases ROS and inflammation, thereby increasing HIF-1α" (ll. 55) and "In contrast, chronic alcohol exposure in adult rats increased HIF-1 mRNA and protein in the brain cortex" (ll93-ll94) need references.

Please reconsider them throughout the manuscript.

In addition, not a small number of citation seems to be incorrect.

For example, in the chapter 2 "The role of HIF-1α in alcohol-mediated brain damage", references [17, 18] are not relevant with "brain". Please reconsider the references throughout the manuscript.

#HIF

"Hypoxia inducible factor" should be #Hypoxia-inducible factor". (ll. 18)

Please also note that HIF-1a is the name of the subunit of the transcription factor HIF-1. Only HIF-1a does not function as a transcription factor.

#iNoS

"iNoS" should be "iNOS"(ll.99).

Author Response

We would like to thank the reviewers for thoroughly reviewing our manuscript.  We have clarified their points of concern which have improved the manuscript. We have addressed the reviewers’ comments as indicated by the following point-by-point responses.  Red font in the manuscript indicates major changes to the original text.

Reviewer #2

Comment 1) “…a chapter of molecular mechanisms of regulation of HIF activity can provide more precise information to potential readers.  Although HIFs are identified as the transcription factor activated in response to hypoxia, they can be activated by various stimulation or stress including inflammation and reactive oxygen species (ROS) by different molecular mechanisms from that of hypoxia.”

Response 1) We have included a section detailing these molecular mechanisms which regulate HIF activity to provide a more comprehensive view into how HIF is regulated (pages 1- 2, lines 38-81).  The section explains how HIF-1α is stabilized by oxygen dependent and oxygen independent mechanisms such as inflammation and ROS stimulation.  Further, the section explains the debate that ROS may in fact reduce HIF-1α stabilization.

Comment 2)” …lacks proper citations throughout the manuscript.  For example, in page 2 ’Alcohol increases ROS and inflammation, thereby increasing HIF-1α’ (ll. 55) and ‘In contrast, chronic alcohol exposure in adult rats increased HIF-1 mRNA and protein in the brain cortex’ (ll93-ll94) need references.  Please reconsider them throughout the manuscript.”

Response 2) We have thoroughly reviewed our manuscript and have added the appropriate citations to statements throughout. 

Comment 3) “…citation seems to be incorrect.  For example, in the chapter 2 ‘The role of HIF-1α in alcohol-mediated brain damage’, references [17, 18] are not relevant with ’brain’. Please reconsider the references throughout the manuscript.”

Response 3) We have thoroughly checked our manuscript for citation errors and have corrected these errors throughout.  The references Biswas, 2013 and Treins, 2005 were included in this section to provide additional evidence of the relationship between HIF-1α and insulin/IGF-1 rather than directly relating to the brain. This statement has been clarified: “Since both insulin and IGF-1 have been shown in other models to stabilize HIF-1α in mouse preadipocytic cells and in human retinal epithelial cells [Biswas, 2013 and Treins, 2005], diminished insulin and IGF following prenatal alcohol exposure may attribute to downregulation of HIF-1α seen in the brain by similar mechanisms.” (page 3, lines 102-105)

Comment 4) “’Hypoxia inducible factor’ should be ’Hypoxia-inducible factor’ (ll. 18).”

Response 4) We have corrected the text to reflect “Hypoxia-inducible factor” (page 1, line 18 and page 1, line 39).

Comment 5) “…HIF-1a is the name of the subunit of the transcription factor HIF-1. Only HIF-1a does not function as a transcription factor.”

Response 5) We have corrected the text to clearly differentiate that HIF-1 is the transcription factor while HIF-1α is a subunit within the protein complex.  The following changes have been made to the text to provide more clarity: (a) “Hypoxia-inducible factor (HIF)-1 is a transcription factor comprised of HIF-1α and HIF-1β subunits that play a critical role in alcohol-mediated organ dysfunction” (page 1, lines 18-20); and (b) “Beyond alcohol affecting transcription factors such as HIF-1, alcohol also modulates the expression of microRNAs” (page 7, lines 328-329).

Comment 6) “’iNoS’ should be ’iNOS’ (ll.99).”

Response 6) We have corrected “iNoS” to “iNOS” (page 3, line 122)

Reviewer 3 Report

The submitted manuscript from Morris & Yeligar, entitled “Role of HIF-1a in alcohol-mediated multiple organ disfunction” aims to provide a comprehensive overview about the link between HIF-1 and alcohol-mediated organ injuries. The topic is interesting but the manuscript has not quite reached its aim, it lacks some key information as well as it contains some inaccuracies. Therefore, it needs a careful and thoughtful revision.

Major comments:

The interplay between ROS and HIF is highly complex and multilayered. Whether ROS stabilize HIF-1alpha under normoxia and/or hypoxia or not, whether ROS regulate HIF-1alpha levels directly or indirectly as well as what are the exact mechanisms behind up-, down- or stable levels of HIF upon ROS are, so far, subject of a long-term debate with no clear consensus yet. For example, number of studies show that effects of ROS (especially H2O2 which is a byproduct of EtOH metabolism) on HIF is strongly concentration-dependent (low concentrations upregulate HIF-1alpha while high concentrations  prevent HIF-1alpha accumulation) and cell type-dependent. Since excessive ROS are hallmark of alcohol-induced organ injury, I strongly recommend the authors to write a paragraph about ROS and HIF in the introduction; it will help the reader to understand why despite the high ROS levels in some cases HIF is upregulated and in other - downregulated.

In some of the described organs the existing data are not clearly presented; it is not clear whether there is a link between alcohol-injury and HIF - is the effect of alcohol on HIF direct or indirect (via what, or is it just a descriptive observation?). The authors should elaborate more, providing clear statements about the results from those studies concerning the axis “alcohol-HIF” especially in brain, lung and adipose tissue. Just one example, p2, lines 68-92-this paragraph is convoluted, difficult to follow and lacks the clear “take-home” message. Some of the statements are counter-intuitive. If ROS and inflammation (produced by EtOH) are involved in HIF stabilization, why then HIF is downregulated in prenatal  alcohol exposure? How exactly the reduced level of antioxidants enzyme (thus leading to excessive ROS) results in reduced HIF? It is not clear whether indeed these studies showed a link between ROS/inflammation and HIF or it is the interpretation of the authors on the data? It seems that these studies underline the important protective function of HIF stabilization in prenatal stage but this is not clearly stated.

In addition, there are general statements throughout the manuscript concerning the upregulation of HIF by ROS, but since this is not the case in all reported studies (even in this manuscript), the authors should summarize more thoughtfully.

Conclusion paragraph is very short, vague and unfocused. The protective vs contributing role of HIF in alcohol-mediated multiple organ dysfunction is not clearly summarized. Treatment options leading mainly to HIF suppression are mentioned but their relevance in the system/organs (where HIF obviously plays a protective role again alcohol-induced organ injury) are not clear. What indeed authors think after going through the literature? The review very much will benefit by clear author opinion.

Minor comments:

p2, line 50-51 -”Hypoxia, enhanced ROS, and inflammation have all been demonstrated to increase transcription and stabilization of HIF-1α under normoxic conditions” - hypoxia and normoxia as conditions are mutually exclusive, please correct the sentence

p.1, line 38 “Excess ROS increases hypoxia inducible factor (12)” I propose authors check more careful reference 12 and whether the data of this study indeed reflect correctly their  statement.

p2, line 47-48 & p3, line 106- “The HIF-1 complex acts as a transcription factor to regulate approximately 60 genes”. HIF targets are more than 60 genes. The authors need to include a relevant reference.

p5, line 243 “ROS stabilizes HIF-1α by decreasing its hydroxylation by prolyl hydroxylases”-this is one of the mechanisms by which ROS work on HIF, but there are also others; please re-write the sentence to imply the existence of other suggested mechanisms as well

p6, line252-253-”Pretreatment with M30 partially reduced apoptosis, ROS levels, and secretion of pro-inflammatory  mediators in hepatocytes” - please specify which are those  pro-inflammatory mediators

Author Response

We would like to thank the reviewers for thoroughly reviewing our manuscript.  We have clarified their points of concern which have improved the manuscript. We have addressed the reviewers’ comments as indicated by the following point-by-point responses.  Red font in the manuscript indicates major changes to the original text.

Reviewer #3

Comment 1) “The interplay between ROS and HIF is highly complex and multilayered… Since excessive ROS are hallmark of alcohol-induced organ injury, I strongly recommend the authors to write a paragraph about ROS and HIF in the introduction; it will help the reader to understand why despite the high ROS levels in some cases HIF is upregulated and in others it is downregulated.”

Response 1) We understand the concerns regarding potential misinterpretation of the relationship between ROS and HIF-1α.  We have included a section detailing the molecular mechanisms which regulate HIF activity and highlight the mechanisms by which ROS regulates HIF (pages 1-2, lines 38-81). We have additionally described the debate surrounding the ROS-HIF issue by presenting some studies in which ROS increases HIF stability and other studies in which ROS downregulates HIF-1α expression.

Comment 2) “…. The authors should elaborate more, providing clear statements about the results from those studies concerning the axis “alcohol-HIF” especially in brain, lung and adipose tissue…lacks the clear “take-home” message…

In addition, there are general statements throughout the manuscript concerning the upregulation of HIF by ROS, but since this is not the case in all reported studies (even in this manuscript), the authors should summarize more thoughtfully.”

Response 2) We have more clearly described the mechanisms by which alcohol impacts expression of HIF-1α in the brain (page 3, lines 116-119), lung (page 4 lines 166-171), and adipose tissue (pages 4-5 lines 194-197).  Additionally, we have included “take home” messages after each section to clearly summarize each section.  Further, we have provided a summary paragraph to more clearly describe the relationship between HIF and ROS in which we describe contrasting studies where ROS has been shown to increase or decrease HIF-1α (pages 1-2, lines 38-81).  If not clearly defined, we have made inferences as to studies’ conclusions; otherwise, we have added citations to provide references to our statements where study conclusions were reached. 

Comment 3) “Conclusion paragraph is very short, vague and unfocused… What indeed do the authors think after going through the literature? The review very much will benefit by clear author opinion.”

Response 3)  We have expanded our conclusion (page 9 , lines 400-431) to: (a) include a figure describing the differential expression of HIF-1α is due to type of alcohol exposure (acute/binge/chronic) and time of alcohol exposure (prenatal vs. adult) (Figure 1), (b) summarize the effects of alcohol on HIF-1 in multiple organ systems, (c) summarize new therapeutic avenues to mitigate the negative effects of alcohol exposure as it relates to alcohol-mediated induction or depression of HIF-1α (Table 1).

Comment 4) “p2, line 50-51 - ’Hypoxia, enhanced ROS, and inflammation have all been demonstrated to increase transcription and stabilization of HIF-1α under normoxic conditions’ - hypoxia and normoxia as conditions are mutually exclusive, please correct the sentence.”

Response 4) We have omitted this sentence to avoid any confusion to the readers. (page 2, lines 66-67). 

Comment 5) “p.1, line 38 “Excess ROS increases hypoxia inducible factor (12)” I propose authors check more careful reference 12 and whether the data of this study indeed reflect correctly their statement.”

Response 5) We have included a section on molecular mechanisms which regulate HIF activity to our manuscript (pages 1-2, lines 38-81) and we have removed the citation in question

Comment 6) “p2, line 47-48 & p3, line 106 - ’The HIF-1 complex acts as a transcription factor to regulate approximately 60 genes’. HIF targets are more than 60 genes. The authors need to include a relevant reference.”

Response 6) We have included an updated a reference for this statement (page 2, line 64).

Comment 7) “p5, line 243 ‘ROS stabilizes HIF-1α by decreasing its hydroxylation by prolyl hydroxylases’ - this is one of the mechanisms by which ROS work on HIF, but there are also others; please re-write the sentence to imply the existence of other suggested mechanisms as well.”

Response 7) We have clarified the statement to, “Some studies have illustrated that ROS has the ability regulate HIF-1α stability, although this is still under debate” (page 6, lines 295-296).

Comment 8) “p6, line252-253 – ‘Pretreatment with M30 partially reduced apoptosis, ROS levels, and secretion of pro-inflammatory mediators in hepatocytes’ - please specify which are those pro-inflammatory mediators.”

Response 8) To provide more clarity, we have changed the sentence to “Pretreatment with M30 partially reduced apoptosis, ROS levels, and secretion of pro-inflammatory mediators (TNF-α, IL-6, IL-1β and IL-18) in hepatocytes” (page 7, lines 305-306).

Round 2

Reviewer 2 Report

 I think that the manuscript has been significantly 

improved and now warrants publication in Biomolecules.

Reviewer 3 Report

The paper is significantly improved; all my concerns are addressed properly